# Engineered Cell Line Imaging Assay Differentiates Pathogenic from Non-Pathogenic Bacteria

**DOI:** 10.3390/pathogens11020209

**Published:** 2022-02-04

**Authors:** Shelby M. B. Phillips, Carson Bergstrom, Brian Walker, George Wang, Trinidad Alfaro, Zachary R. Stromberg, Becky M. Hess

**Affiliations:** Chemical and Biological Signatures Group, Pacific Northwest National Laboratory, Richland, WA 99352, USA; shelby.phillips@pnnl.gov (S.M.B.P.); carson.bergstrom@pnnl.gov (C.B.); brian_m_walker@rush.edu (B.W.); gpswang@student.ubc.ca (G.W.); trinidad.alfaro@pnnl.gov (T.A.); zachary.stromberg@pnnl.gov (Z.R.S.)

**Keywords:** detection, fluorescent reporters, mammalian cell engineering, pathogenicity, protein kinase, bio-agent agnostic signatures

## Abstract

Cell culture systems have greatly expanded our understanding of how bacterial pathogens target signaling pathways to manipulate the host and cause infection. Advances in genetic engineering have allowed for the creation of fluorescent protein readouts within signaling pathways, but these techniques have been underutilized in pathogen biology. Here, we genetically engineered a lung cell line with fluorescent reporters for extracellular signal-related kinase (ERK) and the downstream transcription factor FOS-related antigen 1 (Fra1) and evaluated signaling after inoculation with pathogenic and non-pathogenic bacteria. Cells were inoculated with 100 colony-forming units of *Acinetobacter baylyi*, *Klebsiella pneumoniae*, *Pseudomonas aeruginosa*, *Streptococcus agalactiae*, or *Staphylococcus epidermidis* and imaged in a multi-mode reader. The alamarBlue cell viability assay was used as a reference test and showed that pathogenic *P*. *aeruginosa* induced significant (*p* < 0.05) cell death after 8 h in both wild-type and engineered cell lines compared to non-pathogenic *S*. *epidermidis*. In engineered cells, we found that Fra1 signaling was disrupted in as little as 4 h after inoculation with bacterial pathogens compared to delayed disruption in signaling by non-pathogenic *S*. *epidermidis*. Overall, we demonstrate that low levels of pathogenic versus non-pathogenic bacteria can be rapidly and sensitively screened based on ERK-Fra1 signaling.

## 1. Introduction

The global diversity of identified bacterial species is staggering, considering it is predicted that many species remain undiscovered [1]. Most bacterial species have no known adverse effect on humans, and some strains can even be beneficial [2,3]. However, a fraction of bacterial strains can be pathogenic to humans. Detection methods such as PCR, ELISA, and lateral flow assays are commonly used for bacterial pathogens, but genetic similarity among strains and cross-reactivity issues with probes and antibodies can hamper detection [4]. Therefore, recent advances in matrix-assisted laser desorption/ionization time-of-flight mass spectrometry (MALDI-TOF) [5,6,7], optical phenotyping of bacterial colonies, and fiber optic biosensors [8,9,10] can further detect and discriminate between strains. However, many of these methods are not functional screens and cannot determine whether a pathogen has the necessary factors to colonize and cause disease. Differentiating between a pathogen and a non-pathogen in the laboratory can be challenging as pathogenicity is a dynamic feature between the host and bacterial strain [11]. Evaluating whether a bacterial strain is a pathogen often requires the consideration of multiple factors, including the presence of functional bacterial virulence factors, the ability for the bacterial strain to colonize or invade host cells, and the ability to cause harm [12].

The study of host–microbe interactions has led to several insights into how bacteria cause disease. Classically, the interrogation of infection processes has relied heavily on the experimental evaluation of bacterial interactions with cultured mammalian cell lines [13]. This has led to advances in visualizing host–pathogen interactions [14,15], understanding pathways elicited by pathogens [16], discovery and contribution of virulence factors [17], and the elucidation of host responses [18]. However, experimental evaluation of bacterial interactions with cell lines can be laborious. More recently, high-throughput imaging has made it possible to rapidly screen hundreds to thousands of samples. Still, high-throughput imaging has primarily been used to evaluate compounds with antibacterial potential in pathogen research [19,20]. These techniques could create high-throughput imaging assays to screen for pathogens if distinguishing cellular signals are known.

Bacterial pathogens target pathways that control critical cellular functions such as cytoskeletal dynamics, autophagy, and cellular homeostasis [21,22]. Within a signaling pathway, a pathogen often targets specific host proteins to manipulate cells [23]. Protein kinases are good candidates for pathogens to manipulate because kinase activity has a considerable influence over cellular proteins [24]. For example, Krachler et al. reviewed that pathogens inhibit ERK (extracellular signal-related kinase) from activating downstream host responses such as the transcription factor Fra1 (FOS-related antigen 1) [23]. Bacterial pathogens inhibit Fra1 to suppress the activation of pro-inflammatory responses [25]. The activation and inhibition of such proteins can now be visualized in live cells using genetic engineering tools to create fluorescent reporters [26]. However, these techniques have been underutilized for creating cell-based imaging assays to differentiate pathogens from non-pathogens.

There is a critical need for tools that can rapidly and sensitively differentiate pathogenic from non-pathogenic bacteria without prior knowledge of the strain. In particular, respiratory pathogens are among the top causes of death globally, and the emergence of novel pathogens threatens public health [27,28]. To overcome current shortcomings, we aimed to determine whether an engineered lung cell line with fluorescent readouts for the protein kinase ERK and transcription factor Fra1 was able to differentiate between pathogenic versus non-pathogenic bacteria in a high-throughput imaging-based cell assay.

## 2. Results

### 2.1. Pathogenic Bacteria Induce Significant Cell Death in A549 Cell Lines

The alamarBlue assay is a gold standard assay for determining cell death [29,30]. We subjected A549 wild-type (WT) and A549 ERK-Fra1 cells engineered with fluorescent reporters to 100 colony-forming units (CFU) of the non-pathogenic bacterium *Staphylococcus epidermidis* or pathogenic bacterium *Pseudomonas aeruginosa* for 8 h. We performed the alamarBlue assay following this incubation to determine whether the pathogen was cytotoxic to both A549 cell lines. Overall, we determined that cell viability was significantly (*p* < 0.05) reduced after incubation with *P*. *aeruginosa* compared with *S*. *epidermidis* (Figure 1). In the presence of the non-pathogen, cell survival was 97% in the A549 ERK-Fra1 cell line and 84% in the A549 WT cell line, indicating that this bacterium was not harmful to the mammalian lung cells. Conversely, pathogenic *P. aeruginosa* exposure resulted in 47% cell death in A549 ERK-Fra1 cells and 72% cell death in A549 WT cells, indicating significant pathogenicity in both cell lines.

### 2.2. Imaging Assays

The imaging assays are based on the reporters engineered into the A549 ERK-Fra1 cells. As shown in Figure 2, total ERK abundance was measured by an mCherry (red) signal, and the phosphorylated Fra1 abundance was measured by an mVenus (yellow) signal. Based on previous work showing that pathogens disrupt the ability of ERK to phosphorylate its Fra1 substrate, we hypothesized that the mVenus (yellow) signal would diminish over time in the presence of a pathogen in comparison to a non-pathogenic bacterium [23]. Although we expected that a decrease in Fra1 signaling would be a more rapid indicator for pathogen presence than cell death in the alamarBlue assay, the imaging assay was extended to 12 h in case our hypothesis failed.

### 2.3. Engineered Reporters Have a Strong Fra1 (mVenus) Signal in the Presence of Epidermal Growth Factor (EGF)

To test the engineering strategy for ERK-Fra1, we incubated cells with 10 ng/mL EGF and a 10 μL of LB bacterial growth medium as an untreated control. Imaging assays were conducted over a 12 h time scale to assess ERK-Fra1 activity. As shown in Figure 3, there was strong Fra1 (mVenus) signaling over the full time course with EGF treatment and the control. This was consistent with previous work showing that EGF stimulation occurs within the first few hours of treatment [31].

### 2.4. Non-Pathogenic Bacteria Have Delayed Interruption of ERK-Fra1 Signaling

A strain of *S. epidermidis* was selected as the standard for a non-pathogenic bacterium because it is one of the most abundant colonizers of a healthy human respiratory tract [33]. We tested whether this non-pathogen would impact ERK-Fra1 signaling by inoculating the A549 engineered lung cell line with 100 CFU of log phase bacterial growth. Imaging assays were performed over a 12 h time scale to assess ERK-Fra1 activity for this organism and all following organisms. As shown in Figure 4A, there was no disruption in ERK-Fra1 signaling until the 12 h time point. The Fra1 signal (mVenus) was observed in the nucleus as expected, and the constitutive ERK signal (mCherry) was observed in the cytoplasm.

### 2.5. Engineered Reporters Have Interrupted Fra1 Signaling in the Presence of Pathogens

Next, we tested the effect of pathogens on the ERK-Fra1 signaling pathway. For pathogens, we selected *Acinetobacter baylyi*, *Streptococcus agalactiae*, *Klebsiella pneumoniae*, and the same strain of *P*. *aeruginosa* used previously in the alamarBlue assay. These four organisms were used because they are either common respiratory pathogens (*K*. *pneumoniae* and *P*. *aeruginosa*) [34] or are associated with bacteremia and can spread to infect multiple organs, including the respiratory tract (*A. baylyi* and *S. agalactiae*) [35]. At the 0 h time point, all wells had confluent monolayers, but there was some variability between fluorescence. Variability in fluorescence could be due to cellular metabolism or the state of growth, which may affect Fra1 signaling in the frame of the well that was imaged (Figure 4). However, all cells displayed Fra1 signaling at the 0 h time point. Based on previous reports, we expected to observe reduced Fra1 signaling due to manipulation of kinase signaling within the A549 cell line [23]. All tested pathogens resulted in reduction or disruption of Fra1 signaling within 4 h post-infection (Figure 4B–E). In contrast to non-pathogen exposure, pathogen-infected cells displayed a decrease in Fra1 signal (mVenus) in the nucleus while maintaining the constitutive ERK signal (mCherry) in the cytoplasm. For the opportunistic pathogenic strain of *A*. *baylyi*, reduction of Fra1 signal was observed over the time course, with no Fra1 signaling observed at 12 h. In addition to cells inoculated with *S*. *agalactiae*, *K*. *pneumoniae*, and *P*. *aeruginosa*, there was little to no signal observed for Fra1 (yellow) at 4 h post-infection, and reduction of ERK (red) signaling at 12 h post-infection due to cell death. The last viable time point for *P*. *aeruginosa* was between 11–12 h post-infection due to cell death and rapid bacterial growth leading to loss of ERK (red) signal.

### 2.6. A549 Wild-Type Cells Undergo Damage and Cell Death in the Presence of Pathogens, but Not in the Presence of EGF or Non-Pathogen

In addition to obtaining fluorescence images, we obtained bright field images of the A549 WT cells to assess cell phenotypes over the same 12 h time course. At the 0 h time point, all wells had confluent monolayers, and there was slight variability between wells based on cellular attachment and expansion (Figure 5). As shown in Figure 5A,B, EGF and *S. epidermidis* do not negatively affect cells based on the cell phenotype. For these two treatments, a normal epithelial-like morphology is observed throughout the 12 h incubation and cells remain adherent to the 96-well plate surface, indicating no significant cell death. However, we observed that by 12 h post-infection with pathogenic bacteria, cells were losing adherence to the plate (observed by a rounding phenotype), experiencing loss of tight junctions between cells, and undergoing apoptosis (Figure 5C–F). These data were consistent with the alamarBlue assay indicating that pathogens induce cell death while cells have high viability after incubation with the non-pathogenic *S*. *epidermidis*. In addition, robust bacterial growth was observed for all organisms by 8–12 h post-inoculation that covered the cell monolayer (Figure 5B–F).

## 3. Discussion

Detection of pathogens at low inoculums can be challenging, especially in the context of novel organisms. Here, we developed, applied, and validated an engineered reporter cell line to provide a high-throughput imaging method to screen pathogenic versus non-pathogenic bacteria. Specifically, this system relied upon rapid (minutes) imaging of 96-well plates containing cell lines inoculated with 100 CFU bacteria over a 12 h time course. Mammalian cell-based assays can be used to screen for the presence of viable pathogens in real-time. Common bacterial detection methods such as microarrays [36], PCR [37], and sequencing [38] have been widely adopted to screen for a subset of suspected pathogens. In addition, recent advances in MALDI-TOF MS have led to rapid and accurate detection of bacterial strains at the species, subspecies, and even to the serovar level [5,6,7]. Other methods such as fiber optic biosensors [8,9,10] and optical scattering technology [39,40] can detect pathogens, and some approaches can screen for phenotypes such as whether an antibiotic-induced stress response occurred in bacteria [41]. In contrast, cell-based methods are highly attractive tools for samples that may contain unknown and emerging human pathogens [42]. Unlike molecular methods that may detect dead bacterial organisms [43,44], cell-based assays are functional screens. Probing pathogen interactions with cell cultures has become much less laborious with recent advances in automation, high-throughput imaging, and isolation of single bacterial organisms by microfluidic devices [45,46]. Therefore, the development of cell-based imaging assays are critical for functional screening and elimination of false-negative results from dead, non-hazardous, or non-pathogenic bacteria [47].

Pathogen infection often leads to manipulation and induction of a variety of cell death pathways such as apoptosis, autophagy, necroptosis, pyroptosis, anoikis, and ferroptosis to maintain infection [48,49,50]. To screen for cell health and function after incubation with a bacterial strain, the resazurin reduction alamarBlue assay was used as a reference assay in the current study. We demonstrated that the pathogenic strain of *P*. *aeruginosa* induced significant cell death compared to the non-pathogenic *S*. *epidermidis* after 8 h of incubation with A549 WT and ERK-Fra1 cell lines. Thus, the engineered A549 ERK-Fra1 cells functioned as normal in the reference alamarBlue assay; however, developing screens that can determine pathogenic from non-pathogenic bacteria before the occurrence of cell death would be useful for rapid assessment.

Live-cell imaging can yield high-quality measurements from hundreds of cells in seconds [51]. The usefulness of live-cell measurements relies on a specific signal to measure. Here, we focused on ERK-Fra1 signaling because it is known to be inhibited by several bacterial pathogens [23,52,53]. In addition, established tracking strategies have been developed for ERK activity relying on fluorescent reporters that are convenient, stable, and at low-cost [51]. We confirmed that treatment of engineered cells with EGF induced Fra1 expression as demonstrated previously [54]. Next, we hypothesized that Fra1 signaling in A549 ERK-Fra1 cells would be maintained after incubation with a non-pathogen but interrupted by pathogens. After incubation with a non-pathogenic *S*. *epidermidis*, the A549 ERK-Fra1 cells maintained the Fra1 signal for at least 8 h after inoculation. In contrast, the four pathogens tested (*A*. *baylyi*, *K*. *pneumoniae*, *S*. *agalactiae*, *P*. *aeruginosa*) interrupted the Fra1 signal in A549 ERK-Fra1 cells. Previously, fluorescent reporter systems were primarily limited to creating fluorescent bacterial strains for visualization during infection and assessing pathogen stress responses [55,56,57]. Here, we have developed fluorescent reporters for a eukaryotic cell line that indicate the presence of a pathogen by disruption of the Fra1 signal. To our knowledge, the current study is one of the only studies that has used a reporter cell line to detect the presence of a bacterial pathogen.

There are limitations in our current assessment of the A549 ERK-Fra1 cell line to screen pathogens. We tested a limited number of bacterial strains with only one non-pathogenic strain and four pathogens. Inclusion of a broader set of microbial strains in future studies will more thoroughly evaluate the utility of this system. Moreover, not all pathogens colonize lung cells and cause disease. Thus, the use of reporters would need to be expanded in other cell lines and types such as gastrointestinal and immune cells that pathogens frequently infect [58]. In addition, not all pathogens can manipulate ERK-Fra1 signaling. Therefore, the discovery of novel host signaling pathways impacted by pathogens warrants future study for their use not only in reporter cell lines but for the strategic development of interventions and therapeutics [59].

For cell lines incubated with bacteria or EGF, we also assessed apparent changes to A549 WT cellular phenotype, such as loss of adherence to the 96-well plate and loss of tight junctions between cells. No significant changes were observed after EGF treatment or inoculation with *S*. *epidermidis*. Conversely, inoculation of a pathogen resulted in loss of cellular adherence to the 96-well plate surface, the loss of tight junctions, and the onset of apoptosis. These findings confirm what was observed for ERK-Fra1 signaling in the engineered cell line and help validate the ability of engineered cells to differentiate pathogens from non-pathogens.

Altogether, we show here that pathogenic bacteria disrupted ERK-Fra1 signaling and induced cytopathic effects, whereas the non-pathogenic bacterium did not cause this disruption within 4 h or cytotoxic phenotype. Importantly these findings were observed with a low inoculum of bacteria. Furthermore, the interruption in Fra1 signaling observed (4 h) can potentially serve to shorten the time compared to current cell-based assays for pathogen screening.

## 4. Materials and Methods

### 4.1. Cell Culture

The A549 cell line was obtained from ATCC. Cells were maintained in an incubator with 5% CO_2_, 95% relative humidity, and at 37 °C. Cells were cultured in 1× DMEM (+4.5 g/L glucose, +L-glutamine, and +110 mg/L sodium pyruvate, Gibco #11995-065), 10% fetal bovine serum (Gibco #10438-026), and 1× (100 U/mL) penicillin/streptomycin (Gibco). Cells were passaged every 3 to 4 days at a ratio of 1:4. For cell counting, confluent cells (75–85%) were washed twice with PBS (Gibco), and viable cells were counted using trypan blue staining and a Countess II FL counter (ThermoFisher, Waltham, MA, USA). A549 cell lines were seeded in 96-well plates (Corning #3603 and Greiner Bio-One #655090) for assays.

### 4.2. Engineering A549 Cell to Create A549 ERK-Fra1

The ERK reporter and Fra1 reporter plasmids were kind gifts from Dr. John Albeck (University of California, Davis). The reporter plasmids were sent to Vigene such that the reporter constructs (region between the long terminal repeats of the reporter plasmids) could be cloned into the pLenti-CMV transfer plasmid. The vectors were designed such that the ERK reporter conveyed resistance to geneticin (G418) and the Fra1 reporter conveyed resistance to puromycin. The transfer vectors were then packaged by Vigene into third-generation lentiviruses using envelope and packaging plasmids.

The A549 cell line was transduced using 10^6^ IFU/mL of lentivirus and 10 µg/mL of polybrene (Millipore, Burlington, MA, USA). Briefly, 500 µL of 10^6^ IFU/mL of each of the Fra1 and ERK lentiviruses were added to the well of a 6-well plate, with 1 mL of a solution of 50,0000 cells/mL of A549 cells added to the well. The cells were incubated uninterrupted for 48 h in a CO_2_ incubator. Following incubation, the media was aspirated and replaced with fresh media (described above) containing antibiotics at concentrations determined by using a kill curve. Specifically, 4 µg/mL of puromycin (Invivogen, San Diego, CA, USA) and 1.5 mg/mL of geneticin (G418, Gibco) were used. Cells were transferred to T-25 flasks upon reaching confluency in the 6-well plate and then passaged at a 1:4 ratio every 3 to 4 days.

### 4.3. Flow Cytometry to Enrich for the Stable Reporter Cell Line of A549 ERK-Fra1

Cell populations were enriched by flow cytometry using the BD Influx Fluorescence Activated Cell Sorter (FACS, BD Biosciences, San Jose, CA, USA). Using the 488-nm to trigger cells, samples were analyzed using a 100-μm nozzle. Optimization and calibration of the FACS was performed before each analysis using 3-μm Ultra Rainbow Fluorescent Particles (Spherotech, Lake Forest, IL, USA). Forward and side scatter detectors were used to gate out cellular debris. The 561 nm laser was used to excite mCherry while measuring emission at 585 nm with a 29 nm bandpass filter. Cell populations were selected for enrichment in mCherry signal. Median calculations from 10,000 cells were done using Flow Jo software (Tree Star, Ashland, OR, USA).

### 4.4. Bacterial Culture

All bacterial cultures were obtained from ATCC or the National Institute of Standards and Technology (NIST). The bacterial organism *Staphylococcus epidermidis* (ATCC 14990) and *Klebsiella pneumoniae* (NIST0151) were cultured in nutrient broth (BD Biosciences) at 37 °C with shaking at 200 rpm. *Streptococcus agalactiae* (ATCC 12400) and *Acinetobacter baylyi* (ATCC 33304) were cultured in BHI (BD Biosciences Franklin Lakes, NJ, USA) at 37 °C with shaking at 200 rpm. *Pseudomonas aeruginosa* (ATCC 10145) was cultured in LB (BD Biosciences) at 37 °C with shaking at 200 rpm.

### 4.5. AlamarBlue Assay

The alamarBlue HS assay was conducted according to the manufacturer’s instructions (ThermoFisher #A50101). Briefly, A549 ERK-Fra1 and A549 WT cells were seeded at ~30,000 cells/well in 96-well plates (Corning #3603, Corning, NY, USA) and allowed to adhere to plates for 24 h. Then, the culture medium on mammalian cells was replaced with DMEM supplemented with 1% FBS (cell assay medium) 24 h prior to the challenge assays. Assays were performed in triplicate. Cells were challenged with 100 CFU of the indicated bacterium. Negative controls included 10 μL of bacterial growth medium (LB and BHI); positive controls included 70% ethanol (ThermoFisher). Following the challenge with the bacteria, the media was removed from each well, and the alamarBlue reagent HS diluted 1:10 in cell assay medium was added to each well. A Cytation5 (BioTek Winooski, VT, USA) was utilized to measure fluorescent units for excitation and emission wavelengths (560 and 590 nm, respectively) of alamarBlue over the course of 8 h with readings every 30 min.

### 4.6. Bacterial Challenge Assay with A549 and A549 ERK-Fra1 Cells

Cells were seeded in 96-well plates (Corning #3603 or Greiner Bio-One #655090, Corning NY, USA) at 20,000–30,000 cells/well. The cells were allowed to adhere to the plate for 24 h. Then, the culture medium on mammalian cells was replaced with DMEM supplemented with 1% FBS 24 h prior to the challenge assays to remove antibiotics from the medium. Bacterial cultures were grown overnight as described above. On the day of the challenge assay, the bacteria were subcultured 1:5 in fresh media and allowed to grow for 90 min at 37 °C with shaking at 200 rpm. Next, an optical density reading at 600 nm for each culture was measured, which was used to determine the volume of culture needed to inoculate mammalian cells with 100 CFU of each bacterium. As a positive control for fluorescence of the A549 ERK-Fra1 cells, 10 ng/mL of EGF was used (Peprotech). Negative controls included 10 μL of bacterial growth medium (LB and BHI). All organisms and controls were tested in at least triplicate.

### 4.7. Image Acquisition

Brightfield and fluorescence (mCherry and mVenus) images of A549 WT and ERK-Fra1 cells were obtained using the Cytation5 cell imaging multi-mode reader (Biotek, Winooski, VT, USA). The 96-well plates were maintained at 37 °C with 5% CO_2_ in a BioSpa 8 automated incubator (Biotek). Images were acquired over 12 h using a 20x objective and the Texas Red (586 nm excitation and 647 nm emission) and YFP (500 nm excitation and 542 nm emission) filter cubes (Biotek). Exposure, brightness, and contrast settings were constant across time points for each strain using the Gen5 3.08 software (Biotek).

### 4.8. Image Analysis

Image analysis was conducted using the Gen5 3.08 software package to determine the number of active Fra1 reporters in each culture. In addition to cell count, the cell size (object size, µm), cell area (object area, µm^2^), sum of the area (object sum area, µm^2^), and pixel intensity for each channel (object mean) were assessed in the YFP channel. These data are included in the Appendix A of this paper.

### 4.9. Statistical Analysis

A Student’s *t*-test was used to determine the significance between responses in the alamarBlue assay using GraphPad Prism software. *p* values < 0.05 were considered significant.

## Figures and Tables

**Figure 1 pathogens-11-00209-f001:**
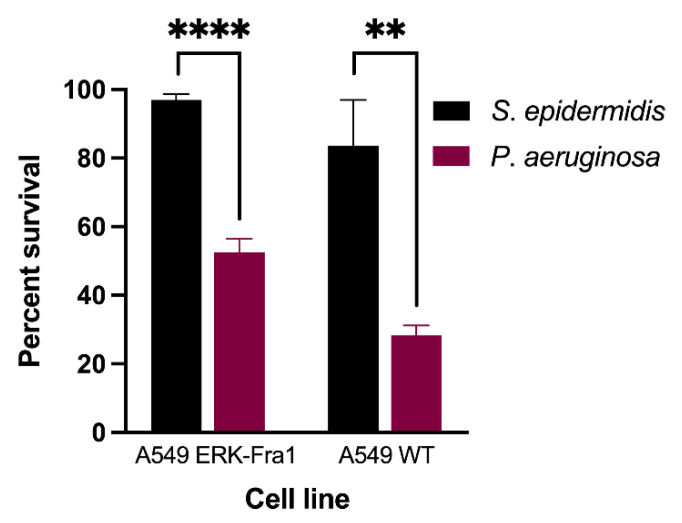
Pathogenic *P. aeruginosa* (red columns) induces significant (*p* < 0.05) cell death in A549 wild-type (WT) and engineered (ERK-Fra1) cells compared to non-pathogenic *S. epidermidis* (black columns) as measured by the alamarBlue assay. The positive control for cell death was 70% ethanol, resulting in less than 10% survival, and the positive controls for viability were LB and BHI medium, resulting in 100% survival. Each column represents the mean of three experiments ± standard deviation. Comparisons were made using a Student’s t-test and asterisks indicate significant differences (** *p* < 0.01, **** *p* < 0.0001).

**Figure 2 pathogens-11-00209-f002:**
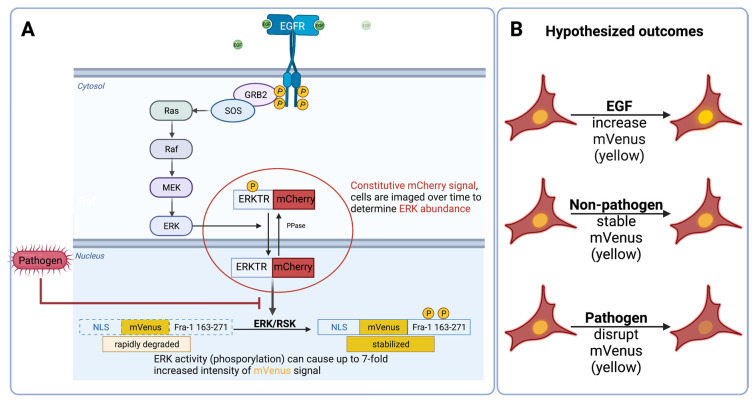
The fluorescent reporter engineering strategy. (**A**) A549 cells were engineered with a mCherry reporter for extracellular signal-related kinase (ERK) and a mVenus reporter for FOS-related antigen 1 (Fra1). (**B**) We hypothesized that in the presence of epidermal growth factor (EGF) or a non-pathogenic bacterium, Fra1 would be phosphorylated and therefore a robust mVenus (yellow) signal would be observed. However, in the presence of a pathogen, we expected disrupted signaling capability for Fra1 and a decrease in mVenus (yellow) signal. Image adapted from Albeck et al. [31] and Sparta et al. [32].

**Figure 3 pathogens-11-00209-f003:**
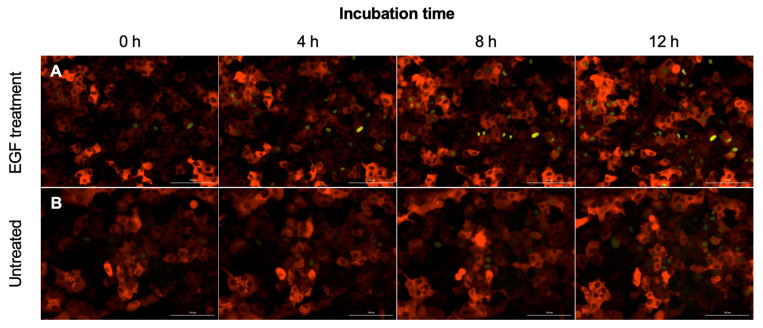
A549 ERK-Fra1 cells in response to (**A**) 10 ng/mL EGF and (**B**) an untreated control (10 μL of LB growth medium) at 0, 4, 8, and 12 h. After EGF treatment, A549 ERK-Fra1 cells have a strong Fra1 signal observed as yellow (mVenus), and for the control a Fra1 signal was observed (yellow, mVenus) across all time points. Each treatment was tested in triplicate. Image analysis of the Fra1 (mVenus) signal for all EGF images can be found in Appendix A. Constitutive ERK signaling is displayed in red (mCherry). Scale bar is 100 µm.

**Figure 4 pathogens-11-00209-f004:**
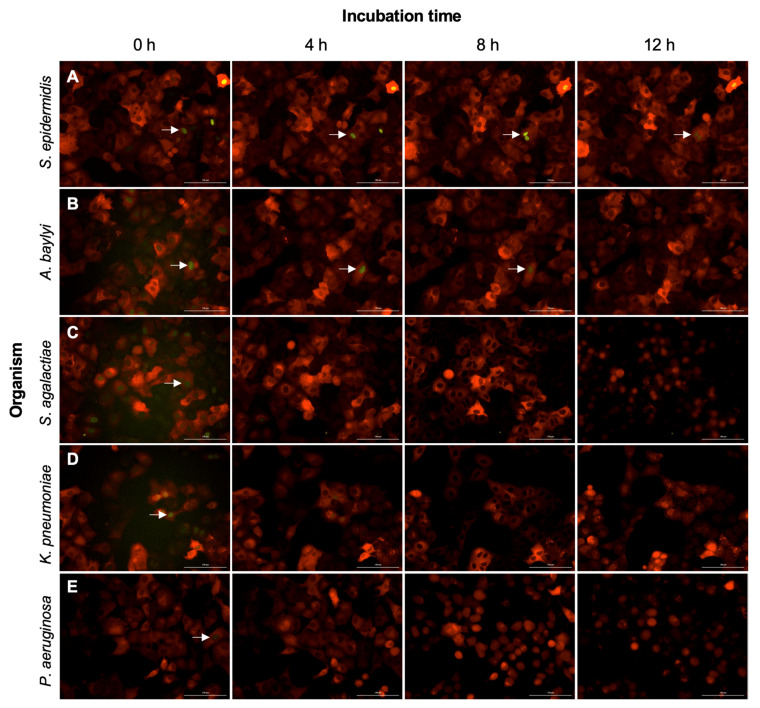
A549 ERK-Fra1 cells in response to (**A**) *S. epidermidis*, (**B**) *A. baylyi*, (**C**) *S. agalactiae*, (**D**) *K. pneumoniae*, and (**E**) *P. aeruginosa* at 0, 4, 8, and 12 h post-infection. Cells have a strong Fra1 signal observed in yellow (mVenus, indicated by arrows) after *S. epidermidis* inoculation (panel **A**). Cells have reduced Fra1 signaling (mVenus) by 4 h after pathogen inoculation (panels **C**–**E**). Each organism was tested in at least triplicate. Image analysis of the Fra1 (mVenus) signal for all images can be found in Appendix A. Constitutive ERK signaling is displayed in red (mCherry). Scale bar is 100 µm.

**Figure 5 pathogens-11-00209-f005:**
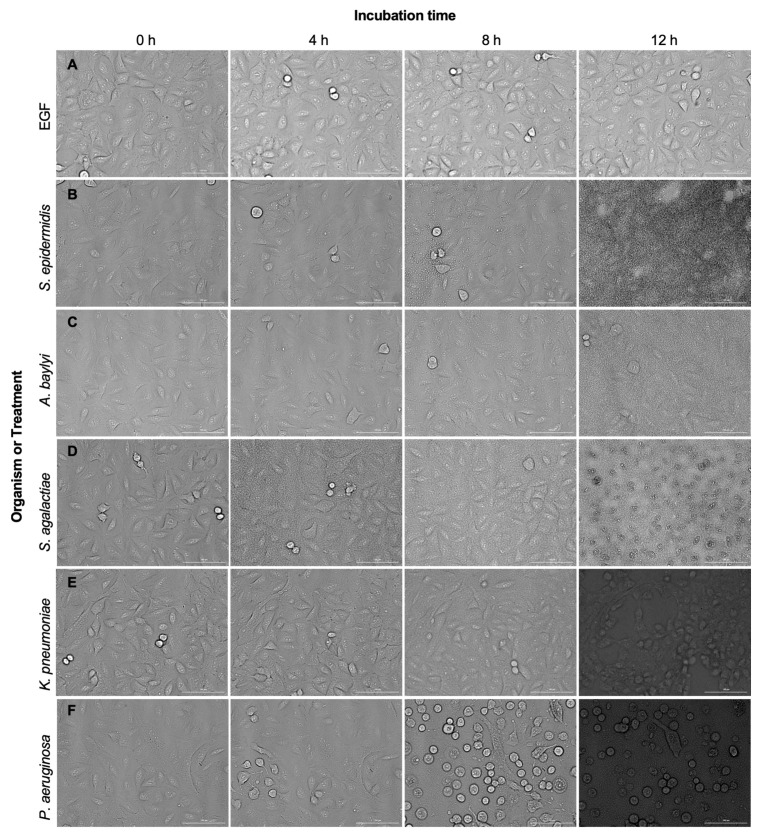
A549 wild-type (WT) cells in response to (**A**) EGF, (**B**) *S. epidermidis*, (**C**) *A. baylyi*, (**D**) *S. agalactiae*, (**E**) *K. pneumoniae*, and (**F**) *P. aeruginosa* at 0, 4, 8, and 12 h post-infection. No significant changes in A549 WT cell phenotype were observed with treatment of EGF or *S. epidermidis* up to 12 h post-infection. Conversely, A549 WT show loss of adherence and tight junctions and the onset of apoptosis in response to the pathogens *A. baylyi*, *S. agalactiae*, *K. pneumoniae,* and *P. aeruginosa*. Each treatment or organism was tested in at least triplicate. Scale bar is 100 µm.

## Data Availability

All data has been made available within the body of this paper and in Appendix A.

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
