# Peer review of "Engineered Cell Line Imaging Assay Differentiates Pathogenic from Non-Pathogenic Bacteria"

_pathogens, 2022, doi:10.3390/pathogens11020209_

Round 1

Reviewer 1 Report

Dear Author/s,

Submitted manuscript Pathogens-1523709 titled as High throughput Imaging assay with Engineered Cell Line Differentiates Pathogenic from Non-Pathogenic Bacteria is a well written draft but have following major issues. Either do the amendments or answer-

  1. The work presented here; whether literally belongs to High throughput imaging assay or simply screening, as highlighted in the title?
  2. Microscopy methods are missing, particularly for high throughput imaging assay and this paper is all about imaging assay.
  3. What is the Object Area, Object Sum Area[Bright Field] Object Mean[Bright Field] as given in the supplementary files? Need to mention in the methods.
  4. If it is feasible, along with representative microscopic images, prepare a graph of using supplementary data and adhere here to make quantitative presentation too.
  5. In figure 1, for % survivability (cell death assay) of A549 cells were only compared with aeruginosa and S. epidermidis, while for imaging assays other strains have also been used. Any specific reason?
  6. What is the green and yellowish color particle/tinge in figure 3 and 4A respectively? Use best quality representation.
  7. Why the incubation time extended till 12 hours if you find the results earlier.
  8. Figure 5 and 6 can be presented in one panel, as they belong to same intervention.
  9. How many A549 cells were incubated with 100CFU of bacteria. In figure 6 apparently (representative images) there is a discrepancy in number due to that. Even after considering the different doubling time.
  10. Various cell death mechanisms have been discussed (Ref # 9-12) but Necroptosis is missing refer to Jondle et al. 2018 PLoS pathogens 14 (10), e1007338 (https://doi.org/10.1371/journal.ppat.1007338)

Overall, this paper needs some attention to make the work crispy, consolidated and reader friendly.

All the Best.

Author Response

Submitted manuscript Pathogens-1523709 titled as High throughput Imaging assay with Engineered Cell Line Differentiates Pathogenic from Non-Pathogenic Bacteria is a well written draft but have following major issues. Either do the amendments or answer-

Response: We thank the reviewer for their diligent review and thoughtful feedback on our manuscript. We have made the following changes to address their comments.

  1. The work presented here; whether literally belongs to High throughput imaging assay or simply screening, as highlighted in the title?

Response: As the reviewer suggested the use of “high throughput” has been removed in the title and replaced by “Engineered Cell Line Imaging Assay Differentiates Pathogenic from Non-Pathogenic Bacteria”. However, the discussion of high-throughput imaging assays remained in the text as this assay is carried out in 96-well plates.

  1. Microscopy methods are missing, particularly for high throughput imaging assay and this paper is all about imaging assay.

Response: A new section (now 4.7) was added to the methods to describe the image acquisition and microscopy methods. Lines 353-360: “Brightfield and fluorescence (mCherry and mVenus) images of A549 WT and ERK-Fra1 cells were obtained using the Cytation5 cell imaging multi-mode reader (Biotek, Winooski, VT). The 96-well plates were maintained at 37°C with 5% CO2 in a BioSpa 8 automated incubator (Biotek). Images were acquired over 12 h using a 20x objective and the Texas Red (586 nm excitation and 647 nm emission) and YFP (500 nm excitation and 542 nm emission) filter cubes (Biotek). Exposure, brightness, and contrast settings were constant across time points for each strain using the Gen5 3.08 software (Biotek).”

  1. What is the Object Area, Object Sum Area[Bright Field] Object Mean[Bright Field] as given in the supplementary files? Need to mention in the methods.

Response: These values in question are from the Gen5 cell analysis software (Biotek). The object size is the size of the cell in µm and the object area is the area of the cells in µm2 determined by multiplying the X-axis pixel length by the Y-axis pixel length. The object sum area[bright field] is the sum of all objects analyzed in the bright field channel. The object mean[bright field] is the sum intensity of all pixels in an object divided by the total number of pixels. This information was added to section 4.8. Line 363-365: “In addition to cell count, the cell size (object size, µm), cell area (object area, µm2), sum of the area (object sum area), and pixel intensity for each channel (object mean) were assessed.”

  1. If it is feasible, along with representative microscopic images, prepare a graph of using supplementary data and adhere here to make quantitative presentation too.

Response: In place of a quantitative graph, we uploaded new microscopic images in Figures 3-5 that have better resolution so that it is easier to observe the change in Fra1 (yellow) signaling.

  1. In figure 1, for % survivability (cell death assay) of A549 cells were only compared with aeruginosaand  epidermidis, while for imaging assays other strains have also been used. Any specific reason?

Response: The alamarBlue assay was used as an established reference assay. Thus, the prototype lung pathogen (P. aeruginosa) and prototype non-pathogen (S. epidermidis) were used as benchmarks to compare with our imaging assay.

  1. What is the green and yellowish color particle/tinge in figure 3 and 4A respectively? Use best quality representation.

Response: New images with better resolution was used for Figures 3-5. The yellow color indicates the Fra1 (mVenus) signaling. We have updated the figure legend to clarify that the yellow color indicates Fra1 signaling in both legends. Line 133-136 (Figure 3): “After EGF treatment, A549 ERK-Fra1 cells have a strong Fra1 signal observed as yellow (mVenus), and for the control a Fra1 signal was observed (yellow, mVenus) across all time points.” Lines 152-153 (Figure 4): “Cells have a strong Fra1 signal observed in yellow (mVenus) after S. epidermidis inoculation (panel A).”

  1. Why the incubation time extended till 12 hours if you find the results earlier.

Response: We hypothesized that the imaging assays with fluorescent reporters would yield results faster than the 8-hour time point of the alamarBlue assay. However, to sufficiently test this hypothesis, we extended the assay to 12 hours in case our hypothesis failed. This rationale was added to the text on Lines 109-111. “Although we expected that a decrease in Fra1 signaling would be a more rapid indicator for pathogen presence than cell death in the alamarBlue assay, the imaging assay was extended to 12 h in case our hypothesis failed.”

  1. Figure 5 and 6 can be presented in one panel, as they belong to same intervention.

Response: Figures 5 and 6 were combined as the reviewer suggested into one figure (now Figure 5).

  1. How many A549 cells were incubated with 100CFU of bacteria. In figure 6 apparently (representative images) there is a discrepancy in number due to that. Even after considering the different doubling time.

Response: The 96-well plates were seeded with ~30,000 cells per well. This information was added to the methods on Lines 339-340: “Cells were seeded in 96-well plates (Corning #3603) at ~30,000 cells/well. The cells were allowed to adhere to the plate for 24 h.”

The cells were allowed to grow and attach to the wells, which may have caused slight variability between wells. At hour 0, the images show a confluent layer of cells with little to no gaps. At later time points in Figure 5 (previously Figure 6), the cells begin to lose adherence, which can impact the number of cells observed. This information was updated in the Results (section 2.6). Lines 184-193: “At the 0 h time point, all wells had confluent monolayers, and there was slight variability between wells based on cellular attachment and expansion (Figure 5). As shown in Figures 5A and 5B, EGF and S. epidermidis do not negatively affect cells based on the cell phenotype. For these two treatments, a normal epithelial-like morphology is observed throughout the 12 h incubation and cells remain adherent to the 96-well plate surface indicating no significant cell death. However, we observed that by 12 h post-infection with pathogenic bacteria, cells were losing adherence to the plate (observed by a rounding phenotype), experiencing loss of tight junctions between cells, and undergoing apoptosis (Figure 5C-5F).”

  1. Various cell death mechanisms have been discussed (Ref # 9-12) but Necroptosis is missing refer to Jondle et al. 2018 PLoS pathogens 14 (10), e1007338 (https://doi.org/10.1371/journal.ppat.1007338)

Response: The discussion has been revised to include necroptosis and the Jondle et al. reference was added (now reference 50). Lines 228-230: “Pathogen infection often leads to manipulation and induction of a variety of cell death pathways such as apoptosis, autophagy, necroptosis, pyroptosis, anoikis, and ferroptosis to maintain infection”

Reviewer 2 Report

The idea presented in the manuscript of the use of an engineered reporter cell line to provide a high-throughput imaging method to study pathogenic and non-pathogenic bacteria is a novel and very interesting approach to bacteria differentiation.  My main complaint is the rather vague description. The authors very often use vague statements that should be discussed or described in more detail. This applies to the results section as well as to the materials and methods section. Most of the results are based on recorded microscopic images; however, their quality in some cases makes it difficult to confirm the authors' statements. Additional comments can be found below.

1)The authors presents novel approach toward bacteria differentiation, however, there is a lack of more general consideration of the recent development is the field of bacteria detection or identification. The authors are describing conventional bacteria detection methods: biochemical assay, or polymerase chain reaction (PCR), however, it is known that owing to the genetic similarity among strains/ serovars, antibodies or nucleic acid probes show cross-reactions limiting the identification of  strains/ serovars. Therefore, the presented description of the state of art is very poor and limited in my opinion as for the review paper. The authors describe well-known and well-established techniques in microbiological diagnostics but did not focus on describing the latest developments in bacterial detection. The authors should present a more recent description of the current advancements in the field of microbiological diagnostics. I am suggesting Authors read the articles related to other recent alternative methods of bacteria detection as:

Mass spectroscopy (MALDI-TOF):

https://doi.org/10.3390/foods10050933

https://doi.org/10.1016/j.foodcont.2020.107188

https://doi.org/10.1371/journal.pone.0040004

Optical phenotyping of bacteria colonies:

https://doi.org/10.1016/j.bios.2015.01.047

https://doi.org/10.1016/j.bios.2020.112761

https://doi.org/10.1117/1.JBO.21.10.107004

https://doi.org/10.1371/journal.pone.0135035

https://doi.org/10.1007/s00253-013-5495-4

https://doi.org/10.1128/mBio.01019-13

https://doi.org/10.1364/OE.22.026312

Optical fibers sensors:

https://doi.org/10.1016/j.foodcont.2015.09.031

https://doi.org/10.3390/s90705810

https://doi.org/10.1016/j.yofte.2018.09.012

https://www.nature.com/articles/s41598-018-35647-2

Authors should describe a more recent state of the art in field of bacteria sensing /detection as well as indicate the advantages of the proposed technique in comparison with other recently developed alternatives (also these indicated above). These methods should be also indicated and described in sections (1,3) devoted to the recent techniques of bacteria detection.

2) The description of the measurement methodology lacks detailed information on the number of samples tested, the repeatability of the measurements, or the control procedures applied. The authors should complete this information.

3) In the manuscript there are a lot of too general notations and statements that should be refined for a wider audience of readers, e.g. line 268, what does such a notation "emission at 585/29nm" mean? Moreover, the description of some sections in Materials and Methods should be extended ( e.g., Image Analysis).

4) There is a lack of information about the optical system used for the registration of bright-field and fluorescence images presented in the manuscript and the registering conditions. This information should be provided.

5) The quality of all microscopic images is very poor, even when these images are magnified, it is difficult to distinguish anything. In addition, the scales bars are completely unreadable. This should be corrected. Moreover, authors should reconsider whether a combination of phase contrast and fluorescence images will not be better for this manuscript purposes?

6) Fig.3- there is no description of all the meaning of the all used colors? What is the meaning of green or yellow stained objects? This information should be explained in text or in figure caption.

7) What kind of assays were used? AlamarBlue HS (high-sensitivity) cell viability reagent or AlamarBlue cell viability reagent because they show different sensitivities. These assays can detect as few as 20 or 50 cells per well. However, based on tables S1-S6, in some cases there were fewer than 20, 10 or even single cells. These numbers are below the assay sensitivity?

8) Why does the fluorescence signal ( see Fig4, first column for 0h) differ so much between pre-infection samples? We had a comparable number of cells in all cases?

9) The statement of lines 124-125: ‘The Fra1 signal (mVenus) was maintained in the nucleus and the constitutive ERK signal (mCherry) was observed in the cytoplasm’ is not clearly demonstrated based on the recent description of Section 2.4. Authors should explain this issue in more detail in relation to the registered images.

10) Also, the next statement (lines 142-144): ‘All tested pathogens resulted in disruption of Fra1 signaling within 4 h after infection (Figure 4B-4E). In contrast to nonpathogen exposure, pathogen-infected cells showed a decrease in Fra1 signal (mVenus) in the nucleus while maintaining the constitutive ERK signal (mCherry) in the cytoplasm ‘is debatable without detailed explanation and better quality images.

11) Lines 149-150: ‘As shown in 149 Figures 5 and 6A, EGF and S. epidermidis do not have any negative effects on cells based on cell phenotype’ -please explain in more detail.

12) Lines 151-155: “However, we observed that by 12 h post-infection with pathogenic bacteria, cells were losing adherence to the plate (observed by a rounding phenotype), experiencing loss of tight junctions between cells, and undergoing apoptosis (Figure 6B- 153 6E). These data were consistent with the alamarBlue assay indicating that cell death is induced by pathogens while cells have high viability after incubation with the non-pathogenic S. epidermidis.”- one more time the quality of some bright field images on Fig.6 Is very poor. Why did the authors not use the DIC or phase-contrast microscopy for this purpose? Why does the image of P. aeruginosa 12 h after infection have such a low intensity? Due to the multiplication of S. agalactiae cells, it is hard to distinguish A549 cells or even their presence 8 hours after infection based on the images presented.

13) According to the Bacterial Challenge Assay with A549 and A549 ERK-Fra1 cells, I wonder how the authors confirmed the lack of antibiotics in the culture medium of mammalian cells? Did the authors have any results for control samples? What was the repeatability of the results obtained? How many series of samples were examined?

14) Why were the fluorescence images for mCherry registered for emission wavelength of 585 nm, if the maximum of emission is around 610 nm?

15) In the supplementary materials (tables S1-S6) there is no information on the size and area units of the objects.

Author Response

The idea presented in the manuscript of the use of an engineered reporter cell line to provide a high-throughput imaging method to study pathogenic and non-pathogenic bacteria is a novel and very interesting approach to bacteria differentiation.  My main complaint is the rather vague description. The authors very often use vague statements that should be discussed or described in more detail. This applies to the results section as well as to the materials and methods section. Most of the results are based on recorded microscopic images; however, their quality in some cases makes it difficult to confirm the authors' statements. Additional comments can be found below.

Response: We thank the reviewer for acknowledging our novel and interesting approach, their thorough review, and thoughtful feedback on our manuscript. We have added detail to the results and methods including a new section for image acquisition in the methods (section 4.7). To improve the quality of the microscopic images, we have uploaded new images for Figures 3-5. The images with K. pneumoniae were lower resolution and thus were replaced with a different pathogen (L. monocytogenes) to achieve higher resolution images as the reviewer requested. We have made the following changes to address their comments as described below.

1)The authors presents novel approach toward bacteria differentiation, however, there is a lack of more general consideration of the recent development is the field of bacteria detection or identification. The authors are describing conventional bacteria detection methods: biochemical assay, or polymerase chain reaction (PCR), however, it is known that owing to the genetic similarity among strains/ serovars, antibodies or nucleic acid probes show cross-reactions limiting the identification of strains/ serovars. Therefore, the presented description of the state of art is very poor and limited in my opinion as for the review paper. The authors describe well-known and well-established techniques in microbiological diagnostics but did not focus on describing the latest developments in bacterial detection. The authors should present a more recent description of the current advancements in the field of microbiological diagnostics. I am suggesting Authors read the articles related to other recent alternative methods of bacteria detection as:

Mass spectroscopy (MALDI-TOF):

https://doi.org/10.3390/foods10050933

https://doi.org/10.1016/j.foodcont.2020.107188

https://doi.org/10.1371/journal.pone.0040004

Optical phenotyping of bacteria colonies:

https://doi.org/10.1016/j.bios.2015.01.047

https://doi.org/10.1016/j.bios.2020.112761

https://doi.org/10.1117/1.JBO.21.10.107004

https://doi.org/10.1371/journal.pone.0135035

https://doi.org/10.1007/s00253-013-5495-4

https://doi.org/10.1128/mBio.01019-13

https://doi.org/10.1364/OE.22.026312

Optical fibers sensors:

https://doi.org/10.1016/j.foodcont.2015.09.031

https://doi.org/10.3390/s90705810

https://doi.org/10.1016/j.yofte.2018.09.012

https://www.nature.com/articles/s41598-018-35647-2

Authors should describe a more recent state of the art in field of bacteria sensing /detection as well as indicate the advantages of the proposed technique in comparison with other recently developed alternatives (also these indicated above). These methods should be also indicated and described in sections (1,3) devoted to the recent techniques of bacteria detection.

Response: Thank you for providing the references to the recent articles on bacterial detection. We have incorporated many of the provided references into the Introduction and Discussion.

The Introduction was revised on Lines 35-42: “Detection methods such as PCR, ELISA, and lateral flow assays are commonly used for bacterial pathogens, but genetic similarity among strains and cross-reactivity issues with probes and antibodies can hamper detection. Therefore, recent advances in matrix-assisted laser desorption/ionization time-of-flight mass spectrometry (MALDI-TOF), optical phenotyping of bacterial colonies, and fiber optic biosensors can further detect and discriminate between strains. However, many of these methods are not functional screens to determine whether a pathogen has the necessary factors to colonize and cause disease.”

The Discussion was revised on Lines 213-219: “Common bacterial detection methods such as microarrays, PCR, and sequencing have been widely adopted to screen for a subset of suspected pathogens. In addition, recent advances in MALDI-TOF MS have led to rapid and accurate detection of bacterial strains at the species, subspecies, and even to the serovar level. Other methods such as fiber optic biosensors and optical scattering technology can detect pathogens, and some approaches can screen for phenotypes such as whether an antibiotic-induced stress response occurred in bacteria.”

2) The description of the measurement methodology lacks detailed information on the number of samples tested, the repeatability of the measurements, or the control procedures applied. The authors should complete this information.

Response: All assays were performed in at least triplicate and representative images were displayed for each organism or control. For activation of the ERK-Fra1 pathway, EGF treatment was used and is included in Figures 3 and 5. As negative controls, 10 ml of LB or BHI was added to the 96-well plates to represent untreated controls (Figure 3). This information was added to the methods and results.

3) In the manuscript there are a lot of too general notations and statements that should be refined for a wider audience of readers, e.g. line 268, what does such a notation "emission at 585/29nm" mean? Moreover, the description of some sections in Materials and Methods should be extended ( e.g., Image Analysis).

Response: More detail was added throughout the manuscript, particularly in the Methods and Results sections. Line 268 (now Line 313-314) was updated to explain that “The 561 nm laser was used to excite mCherry while measuring emission at 585 nm with a 29 nm bandpass filter.” In the methods, the image acquisition methods were revised, and a new section (now 4.7) was added to the methods to describe the image acquisition and microscopy methods.

4) There is a lack of information about the optical system used for the registration of bright-field and fluorescence images presented in the manuscript and the registering conditions. This information should be provided.

Response: The images were acquired using a Cytation 5 cell imaging multi-mode reader from Biotek. A new section (now 4.7) was added to the methods to describe the image acquisition and microscopy methods. Lines 353-360: “Brightfield and fluorescence (mCherry and mVenus) images of A549 WT and ERK-Fra1 cells were obtained using the Cytation5 cell imaging multi-mode reader (Biotek, Winooski, VT). The 96-well plates were maintained at 37°C with 5% CO2 in a BioSpa 8 automated incubator (Biotek). Images were acquired over 12 h using a 20x objective and the Texas Red (586 nm excitation and 647 nm emission) and YFP (500 nm excitation and 542 nm emission) filter cubes (Biotek). Exposure, brightness, and contrast settings were constant across time points for each strain using the Gen5 3.08 software (Biotek).”

5) The quality of all microscopic images is very poor, even when these images are magnified, it is difficult to distinguish anything. In addition, the scales bars are completely unreadable. This should be corrected. Moreover, authors should reconsider whether a combination of phase contrast and fluorescence images will not be better for this manuscript purposes?

Response: New microscopic images were uploaded for Figures 3-5 for better resolution. When the images are magnified the scale bar is readable and displays “100 mm”. As a reference, we have included that the “Scale bar is 100 mm” in each figure legend. The reviewer is correct that phase contrast microscopy can be used to observe cell morphology and death. However, we respectively think that cellular morphology, cell death, and bacterial organisms can be clearly observed with the new uploaded brightfield images in Figure 5.

6) Fig.3- there is no description of all the meaning of the all used colors? What is the meaning of green or yellow stained objects? This information should be explained in text or in figure caption.

Response: The strategy used for the engineered cell lines is explained in section 2.2 and in Figure 2. We have revised this section for clarity, added a panel B to Figure 2, and added what each color indicates in the figure legends for Figures 3 and 4 that used the fluorescent cell lines.

7) What kind of assays were used? AlamarBlue HS (high-sensitivity) cell viability reagent or AlamarBlue cell viability reagent because they show different sensitivities. These assays can detect as few as 20 or 50 cells per well. However, based on tables S1-S6, in some cases there were fewer than 20, 10 or even single cells. These numbers are below the assay sensitivity?

Response: The almarBlue HS cell viability reagent (ThermoFisher, #A50101) was used in these experiments. The number of cells listed in the supplemental tables represents the number of cells in the frame of the 20x objective in the fluorescence images. Thus, that count is only a fraction of the total number of cells in the well. In total, ~30,000 cells/well were used, which would be within the assay sensitivity. Lines 326-335 were updated to reflect this information: “The alamarBlue HS assay was conducted according to the manufacturer’s instructions (ThermoFisher #A50101). Briefly, A549 ERK-Fra1 and A549 WT cells were seeded at ~30,000 cells/well in 96-well plates (Corning #3603) and allowed to adhere to plates for 24 h. Then, the culture medium on mammalian cells was replaced with DMEM supplemented with 1% FBS (cell assay medium) 24 h prior to the challenge assays. Assays were performed in triplicate. Cells were challenged with 100 CFU of the indicated bacterium. Negative controls included 10 ml of bacterial growth medium (LB and BHI); positive controls included 70% ethanol (ThermoFisher). Following the challenge with the bacteria, the media was removed from each well, and the alamarBlue reagent HS diluted 1:10 in cell assay medium was added to each well.”

8) Why does the fluorescence signal ( see Fig4, first column for 0h) differ so much between pre-infection samples? We had a comparable number of cells in all cases?

Response: A comparable number of cells (~30,000 cells per well) were seeded in each well. The differences in fluorescence signal between samples could be due to the variability between wells based on how cells attached, expanded, or were growing during cell culture. We have revised the manuscript (Lines 165-169): “At the 0 h time point, all wells had confluent monolayers, but there was some variability between fluorescence. Variability in fluorescence could be due to cellular metabolism or the state of growth, which may affect Fra1 signaling in the frame of the well that was imaged (Figure 5). However, all cells displayed Fra1 signaling at the 0 h time point.”

9) The statement of lines 124-125: ‘The Fra1 signal (mVenus) was maintained in the nucleus and the constitutive ERK signal (mCherry) was observed in the cytoplasm’ is not clearly demonstrated based on the recent description of Section 2.4. Authors should explain this issue in more detail in relation to the registered images.

Response: To clarify, we have updated the images in Figures 3 and 4 to demonstrate the mVenus (yellow) signal in the nucleus of cells and mCherry (red) signaling in the cytoplasm. We also uploaded a new panel B in Figure 2 to explain in more detail the engineering strategy and how to interpret the images.

10) Also, the next statement (lines 142-144): ‘All tested pathogens resulted in disruption of Fra1 signaling within 4 h after infection (Figure 4B-4E). In contrast to nonpathogen exposure, pathogen-infected cells showed a decrease in Fra1 signal (mVenus) in the nucleus while maintaining the constitutive ERK signal (mCherry) in the cytoplasm ‘is debatable without detailed explanation and better quality images.

Response: We have uploaded new images for Figure 4 with better resolution.

11) Lines 149-150: ‘As shown in 149 Figures 5 and 6A, EGF and S. epidermidis do not have any negative effects on cells based on cell phenotype’ -please explain in more detail.

Response: As the reviewer has suggested, we have added detail in the results for the EGF and S. epidermidis treatments. Lines 184-190: “At the 0 h time point, all wells had confluent monolayers, and there was slight variability between wells based on cellular attachment and expansion (Figure 5). As shown in Figures 5A and 5B, EGF and S. epidermidis do not negatively affect cells based on the cell phenotype. For these two treatments, a normal epithelial-like morphology is observed throughout the 12 h incubation and cells remain adherent to the 96-well plate surface indicating no significant cell death.”.

12) Lines 151-155: “However, we observed that by 12 h post-infection with pathogenic bacteria, cells were losing adherence to the plate (observed by a rounding phenotype), experiencing loss of tight junctions between cells, and undergoing apoptosis (Figure 6B- 153 6E). These data were consistent with the alamarBlue assay indicating that cell death is induced by pathogens while cells have high viability after incubation with the non-pathogenic S. epidermidis.”- one more time the quality of some bright field images on Fig.6 Is very poor. Why did the authors not use the DIC or phase-contrast microscopy for this purpose? Why does the image of P. aeruginosa 12 h after infection have such a low intensity? Due to the multiplication of S. agalactiae cells, it is hard to distinguish A549 cells or even their presence 8 hours after infection based on the images presented.

Response: We have uploaded new images with better resolution for the brightfield images (now Figure 5). By 12 h post-infection, we observed robust bacterial growth for all organisms. We have added that statement into the manuscript. Due to robust growth and cell death, the intensity of the images can decrease as the cytation5 multi-mode automated reader is focusing on the dead cells floating in the media and no longer on the monolayer.

13) According to the Bacterial Challenge Assay with A549 and A549 ERK-Fra1 cells, I wonder how the authors confirmed the lack of antibiotics in the culture medium of mammalian cells? Did the authors have any results for control samples? What was the repeatability of the results obtained? How many series of samples were examined?

Response: For engineered cells, puromycin and geneticin were used for stable cell transfection and wild-type cells were cultured with pen/strep. To remove antibiotics, the cell culture medium was replaced with cell assay media lacking antibiotics. To confirm the lack of residual antibiotics, we observed robust bacterial growth in the bright field channel in both cell types at 12 h post-infection for all strains (See Figure 5 B-F for wild-types).

At least 3 replicates were used for each organism, EGF treatment, and controls. Representative images were used, which displayed similar results across replicates. EGF was used as a control for ERK-Fra1 signaling and 10 ml of LB bacterial growth medium was used as a negative control.

14) Why were the fluorescence images for mCherry registered for emission wavelength of 585 nm, if the maximum of emission is around 610 nm?

Response: Thank you for this question. The cells were sorted by FACS using an emission wavelength of 585 nm which is within the emission range of 550-650 nm for mCherry. The specific wavelength was dictated by the BD Influx FACS instrument (see Lines 308-316) and is a standard emission for sorting in the red channel to confirm cells were engineered with fluorescent reporters.

For fluorescence images for mCherry, a Texas Red filter cube in the Cytation5 (Biotek) was used with an emission wavelength of 647 nm, which is also in the emission range for mCherry. The Methods have been updated on Lines 353-358: “Brightfield and fluorescence (mCherry and mVenus) images of A549 WT and ERK-Fra1 cells were obtained using the Cytation5 cell imaging multi-mode reader (Biotek, Winooski, VT). The 96-well plates were maintained at 37°C with 5% CO2 in a BioSpa 8 automated incubator (Biotek). Images were acquired over 12 h using a 20x objective and the Texas Red (586 nm excitation and 647 nm emission) and YFP (500 nm excitation and 542 nm emission) filter cubes (Biotek).”

15) In the supplementary materials (tables S1-S6) there is no information on the size and area units of the objects.

Response: These values are from the Gen5 cell analysis software (Biotek). The object size is the size of the cell in µm and the object area is the area of the cells in µm2 determined by multiplying the X-axis pixel length by the Y-axis pixel length. The object sum area[bright field] is the sum of all objects analyzed in the bright field channel. The object mean[bright field] is the sum intensity of all pixels in an object divided by the total number of pixels. This information was added to section 4.8. Line 363-365: “In addition to cell count, the cell size (object size, µm), cell area (object area, µm2), sum of the area (object sum area), and pixel intensity for each channel (object mean) were assessed.”

Round 2

Reviewer 1 Report

In submitted revised version, author/s has made the substantial changes in the manuscript. I appreciate that they have incorporated/answered most on the reviewer’s comments and accommodated the suggestions in the new version. However, I found that images are of this version is worse than the before.  In figure four green channel is also open.

I will emphasize that, please take care of these images more attentively, as images are the backbone of your whole article nonetheless main objectives of the manuscript related to your imaging work. If they are not convincing enough to your statements, it would not be worthy. Resolve this issue.

All the Best.

Author Response

Response: We thank the reviewer for their thoughtful feedback and recognizing the substantial changes to the revised version of our manuscript. To address the comments on images, we have made the following changes as detailed below.

We have uploaded new images for Figures 4 and 5. Arrows were added to Figure 4 to indicate Fra1 signaling for easier interpretation. To obtain higher quality images, we have repeated experiments with a lower seeding concentration of cells (20,000 cells/well) and with additional 96-well plate for optical imaging. It should be noted that this study assessed live cell imaging, and thus it is a dynamic system. During replication of experiments, we have added back the standard lung pathogen (K. pneumoniae) that was used in the original submission. In addition, we have updated the Methods and Results to reflect the new images used in Figures 4 and 5.

Reviewer 2 Report

I would like to thank the authors for the clarifications and corrections made to the manuscript; they are sufficient. 

Author Response

(The authors gave the same response as above.)
